Preparation of transferrin-targeted temozolomide nano-micelles and their anti-glioma effect

Yu Jun
Xue Yonghua Yonghuaxue1980@hotmail.com
Department of Neurosurgery, Putuo Hospital, Shanghai University of Traditional Chinese Medicine , Shanghai , China
Lu Tao
Electronic publication date: 2024 Sep 13
Publication date: 2024
Volume: 12
Electronic Location ID: e17979
Received 2024 Jun 20; Accepted 2024 Aug 6
Copyright: © 2024 Yu and Xue
Copyright year: 2024
Copyright holder: Yu and Xue
License: This is an open access article distributed under the terms of the Creative Commons Attribution License, which permits unrestricted use, distribution, reproduction and adaptation in any medium and for any purpose provided that it is properly attributed. For attribution, the original author(s), title, publication source (PeerJ) and either DOI or URL of the article must be cited.
License URL: https://creativecommons.org/licenses/by/4.0/

Keywords: TMZ, TRF nano-micelles, Glioma

Funding: The authors received no funding for this work.

==============================
Objectives

This study aims to develop brain-targeted temozolomide (TMZ) nanograins using the biodegradable polymer material PEG-PLA as a carrier. The model drug TMZ was encapsulated within the polymer using targeted nanotechnology. Key characteristics such as appearance, particle size, size distribution, drug loading capacity, in vitro release rate, stability, and anti-tumor effects were systematically evaluated through in vitro experiments.

Methods

Transmission electron microscopy (TEM) and Malvern size analyzer were employed to observe the morphological and particle size features of the TMZ nanospheres at various time points to assess stability. The effects of TMZ nanograins on glioma cell viability and apoptosis were evaluated using MTT assays and flow cytometry.

Results

The targeted TMZ nano-micelles were successfully synthesized. After loading and targeted modifications, the particle size increased from 50.7 to 190 nm, indicating successful encapsulation of TMZ. The average particle size of the nano-micelles remained stable around 145 ± 10 nm at 1 day, 15 days, and 30 days post-preparation. The release rate of the nano-micelles was monitored at 2 h, 12 h, 24 h, and 48 h post-dialysis, ultimately reaching 95.8%. Compared to TMZ alone, the TMZ-loaded PEG-PLA nano-micelles exhibited enhanced cytotoxicity and apoptosis in glioma cells. This was accompanied by increased mitochondrial membrane potential and reactive oxygen species (ROS) levels following treatment with the TMZ nano-micelles.

Conclusions

TMZ-loaded nano-micelles demonstrated a gradual release profile and significantly enhanced inhibitory effects on human glioma U251 cells compared to TMZ alone. The findings suggest that TMZ-loaded PEG-PLA nano-micelles may offer a more effective therapeutic approach for glioma treatment.

Introduction

Glioblastoma multiforme (GBM) is a highly heterogeneous histological type of tumor in the central nervous system (CNS), accounting for approximately 81% of CNS malignancies (Xu et al., 2020). It is regarded as the most common and malignant brain tumor, accompanied with 5.6% 5-year survival rate and 12 to 15 months overall survival (Pant & Lim, 2023). The main treatment options include surgical resection, radiotherapy, chemotherapy, targeted therapy, and symptomatic treatment (Yalamarty et al., 2023). Surgical resection is one of the main methods for treating glioblastoma (Smith et al., 2022). Radiotherapy is a common treatment for glioblastoma that uses radiation to kill cancer cells and control or alleviate symptoms (Jeon et al., 2023). Chemotherapy aims to kill cancer cells with drugs and has a certain effect on treating glioblastoma (Omuro & DeAngelis, 2013). Chemotherapy is often combined with surgery and radiotherapy to reduce tumor size and prolong survival. However, due to strong invasive nature and high genetic heterogeneity, it is unsatisfied about the outcomes (Parr et al., 2020). Additionally, the blood-brain barrier limits the transport and diffusion of chemotherapy drugs from the blood to the brain, reducing the ability of chemotherapy drugs to reach tumor sites, leading to poor prognosis of human gliomas, with a median overall survival of 14.6 months (Karlsson et al., 2021; Wang et al., 2021a).

Temozolomide (TMZ) is an orally administered alkylating agent that exerts anti-tumor effects by inducing cell death through DNA methylation (Li et al., 2023a). Due to its good tolerance, high permeability across the blood-brain barrier, rapid elimination and relatively short half-life, as well as oral bioavailability of up to 100%, TMZ is a first-line drug for treating human glioma (Chakravarti et al., 2006). This quick clearance from the body results in its high initial bioavailability but reduces its overall efficacy in treating GBM over prolonged periods. Although TMZ is a standard drug for glioma treatment, its efficacy is unsatisfactory and its effect on gliomas can only be maintained for few months (Iturrioz-Rodríguez, Sampron & Matheu, 2023). As a result, it is necessary to discover other feasible methods to further enhance the efficacy of TMZ.

In recent years, drug carriers such as microspheres, microcapsules, liposomes, and polymer nanoparticles have been playing an increasingly significant role as effective delivery tools in the development of novel cancer therapies (Li et al., 2023b; Mainini & Eccles, 2020). On the one hand, they can improve the solubility and stability of drugs, extending the circulation time of drugs in the body. On the other hand, they can target drug delivery to tumor tissues, improving the intra-body distribution of drugs, effectively reducing drug toxicity, and enhancing bioavailability (Javia et al., 2022; Zielińska et al., 2020). Among these, the development and application of liposomes and polymer nanoparticles are the most widespread (Ghezzi et al., 2021). Currently, a series of chemically synthesized polymers have been developed and applied in the preparation of drug carriers. Poly lactic acid (PLA), a polyester polymer material, is non-toxic and biodegradable. It has been approved by the United States Food and Drug Administration (FDA) for medical use, making it a highly promising polymer carrier material (Nofar et al., 2019). Poly ethylene glycol (PEG) is a polymer material obtained by ring-opening polymerization (ROP) of ethylene oxide. It has good solubility (soluble in water and conventional organic solvents), biocompatibility, modifiability, and can evade recognition by the human immune system, possessing a property of remaining “invisible” in the body (Zhao et al., 2020). Transferrin (TRF) is frequently utilized as a ligand for active targeting due to its high interspecies receptor binding equivalence and the overexpression of its receptor in tumor cells (Foglietta et al., 2023; Wu et al., 2022). The polymer material PEG-PLA, obtained by ROP of PLA using PEG as an initiator, possesses amphiphilic structure, good biocompatibility, and biodegradability. The copolymerization of PLA and PEG can enhance drug loading capacity, minimize the burst release effect, extend the in vivo residence time of drugs, and prevent their engulfment by macrophages (Xiao et al., 2010). This type of material serves as an excellent drug carrier (Wang et al., 2021b).

Herein, we hypothesized that the encapsulation of TMZ by PEG-PLA allows for sustained release, enhancing the anti-tumor efficacy in vitro.

Materials and Methods

Synthesis and characterization of nanomedicines

Mix NH2-PEG-PLA (3 mg/mL) and TMZ (2 mg/mL) in a 1:1 mass ratio, dissolve synthesize copolymer in DMF solution, and stir slowly at room temperature in the dark for 2 h. Then the mixture was transferred to a 3.5 kDa dialysis bag for overnight dialysis in 18.2 MΩ deionized water to obtain nano particles loaded with TMZ. Change the water every 4 h to remove DMF, thereby obtaining the self-assembled conjugate. After drug loading, the volume increased from 2 mL to 3.06 mL. The measured concentration is 152.825 μg/mL, and the drug loading capacity is calculated as follows: volume × concentration/2,000. After finishing the previous steps, it is essential to characterize the loaded nanoparticles for size and stability. According to the instructions, TRF was conjugated to the surface of nanoparticles via PEG, PLA, thereafter the morphology and size of nanoparticles were characterized through transmission electron microscope (TEM) and Malvern particle size analyzer. The diagram of TPPT preparation and drug loading were shown in Fig. 1.

Figure 1 Diagram of nano-micelles preparation.

(A) Self assembly of polymer material PEG-PLA. (B) TMZ and TRF loading with PEG-PLA.

In order to investigate the morphological characteristics of targeted nano carriers and targeted nano drugs, prepared sample was dropped onto a copper grid for 2 min, then the leftover solution was quickly blotted away with filter paper, copper grid was air-dried. Further procedure was operated and observed under the guidance of TEM. Moreover, the nanoparticle size and zeta potential were tested according to the Malvern Particle Size Analyzer instructions. Besides, the stability of targeted nano carrier was evaluated by comparison with the particle size in day 1, day 15 and day 30.

Drug loading and release performance of targeted nano carrier

Nano drug was placed in a dialysis bag which then was immersed in a PBS solution. A total of 50 μl dialysate was collected at 2 h, 12 h, 24 h, and 48 h, then equal volume of acetonitrile was added into the collected dialysate. After that, the mixed solution was 5-fold diluted with 0.1 M hydrochloric acid. Release performance of nano carrier was calculated based on the concentration of TMZ at different time points by HPLC.

Cell culture

Human GBM cells U251, obtained from American type culture collection (ATCC, Manassas, VA, USA), were cultured with 10% fetal calf serum (Fisher Scientific, Waltham, MA, USA), 1% penifilin/streptomycin (Life Technologies, Grand Island, NY, USA) in Dulbecco’s modified Eagle’s medium (DMEM, Invitrogen, Carlsbad, CA, USA). Cells were incubated at 37 °C in a humidified incubator (SANYO, Moriguichi, Osaka, Japan) containing 5% CO2.

Cell viability

Cell viability was conducted by MTT reagent (C0009S; Beyotime, Shanghai, China) under the guidance of manufacture’s instructions. Briefly, U251 cells were seeded into a 96-well plate (1 × 105 cells/well) and were incubated for 24 h. Cells were treated with various concentration of TMZ to depict a Cells were then treated with PEG-PLA micelle (PP), TMZ, PEG-PLA-TMZ micelle (PPT), TRF-loaded PPT (TPPT) for another 24 h, and 20 μL of MTT (5 mg/mL) was added to each well and continue to incubate for 4 h. Formed formazan crystals were dissolved in DMSO, and cell viability was calculated according to absorbance at 570 nm as measured using a microplate reader (BioTek, Winooski, VT, USA).

Mitochondrial membrane potential

Mitochondrial membrane potential was assessed via the fluorescence intensity under the fluorescence microscope (Leica, Wetzlar, Germany). Briefly, U251 cells were seeded in 6-well plate (105 cells/well) and incubated overnight. Then cells were treated with TMZ, PPT and TPPT for 24 h. Subsequently, 1 mL culture media and equal volume of JC-1 working solution (C2006; Beyotime, Shanghai, China) were added to incubate for 20 min after being washed with PBS twice. After removing the stained solution, cells were washed twice with buffer solution and then replaced with 1 mL culture medium. Finally, red and green fluorescence were observed separately.

Detection of reactive oxygen species

Reactive oxygen species (ROS) levels were determined according to the instructions of ROS detection kit (S0033M; Beyotime, Shanghai, China). Briefly, logarithmic phase U251 cells were seeded at a concentration of 1 × 105 cells per well in a 6-well plate. After overnight incubation, cells were further treated with TMZ, PPT and TPPT, respectively for 24 h. After discarding the culture medium, cells were washed twice with PBS. Then DCFH-DA probe was diluted 1,000 times in serum-free medium. A total of 1 mL of the diluted probe was added to each well and incubated for another 20 min. The plate was gently inverted every 3–5 min to ensure thorough contact between the probe and the cells. After incubation, cells were washed three times with serum-free medium to remove excess probe that did not enter cells. Finally, ROS levels were evaluated with the detected optical density (OD) value at an excitation wavelength of 488 nm and an emission wavelength of 525 nm.

Flow cytometry analysis

Logarithmic-phase human U251 cells were seeded at a concentration of 1 × 105 cells/well in a 6-well plate overnight. Then cells were treated with 1 μM TMZ, PPT and TPPT for 48 h. Cells were collected after discarding the supernatant through centrifuge at 1,000 g for 5 min. Then 5,000 to 10,000 resuspended cells were taken to remove the supernatant by centrifuge. Resuspend cells in the mixture of 195 μL Annexin V-FITC binding buffer, 5 μL Annexin V-FITC and 10 μL propidium iodide staining solution. Incubate the cells in the dark at 20–25 °C for 20 min. Apoptosis status was finally analyzed using flow cytometry (BD Biosciences, Franklin Lakes, NJ, USA).

Statistical analysis

The data are presented as the means ± standard deviation. Mean differences were evaluated using non-parametric Kruskal–Wallis test. A p value lower than 0.05 was considered statistically significant.

Results

Preparation and characterization of TMZ-incorporated PEG-PLA micelles

To preliminarily confirm the success of the copolymer, the morphology and size of TPPT were captured through TEM. It was observed that TPPT micelles enlarged compared to PP, indicating the loading of TMZ drug component inside PEG-PLA (Fig. 2A). To further verify the changes in nanoparticle size, this study utilized a nanoparticle size analyzer to measure the particle size changes of PP, PPT, and TPPT. As shown in Fig. 2B, it can be seen that the nanoparticle size increased from 50.7 to 141.7 nm before and after drug loading, indicating the successful loading of TMZ into the nanoparticles. Furthermore, after targeted modification, the particle size increased to 190 nm, confirming the success of the targeted modification. To further demonstrate the copolymer under certain conditions, stability experiments were conducted. As is shown in Fig. 3, the results demonstrated that there is no significant difference in particle size of TPPT between different timepoints, which indicates that the TPPT is relatively stable. Meanwhile, we also checked the concentration of TMZ release from TPPT micelles over time, it shows that drug content increases from 0% to 95.8% within 48 h, which indicates that sustained release of TMZ from TPPT is achieved.

Figure 2 Characterization of TMZ-incorporated PEG-PLA micelles.

(A) The morphology of the copolymer captured by scanning electron microscopy. (B) The particle size of the copolymer measured by a nanoparticle size analyzer.

Figure 3 Stability and sustained release effects of TPPT copolymers.

(A) The particle size of TPPT measured at different time points (1 d, 15 d, 30 d) using a nanoparticle size analyzer. (B) Determination of drug concentration of TPPT at different time points.

Effect of copolymers on the cell viability of U251

Further research on PEG-PLA nanomaterials was conducted to evaluate their potential therapeutic effects. MTT viability assay was employed to compare the cytotoxicity of drug-loaded micelles and free TMZ on U251 cells in vitro. We first performed the MTT assay to detect the dose response survival curve of TMZ on U251 cell. The half maximal inhibitory concentration (IC50) was around 500 μM (Fig 4A). Then we administered the same dose of TMZ, PP, PPT, TPPT, and vehicle control to compare their influences on cell viability. The results indicated that PEG-PLA itself did not significantly affect cell viability. However, after loading TMZ, it significantly inhibited the growth of U251 cells, and the activity was further enhanced after loading TRF (Fig. 4B).

Figure 4 Analysis of the activity of nanoparticle micelles vs free TMZ on U251 cells.

(A) 1 × 105 cells/well were seeded in a 96-well plate were treated with PP, TMZ, PPT and TPPT for 48 h, and an MTT assay was used to assess cell viability (n = 5). (B) 1 × 105 cells/well were seeded in a 96-well plate were treated with various concentration of TMZ and survival rate was quantified compared with control group. Values are presented as the means ± standard deviation (n = 5). Kruskal–Wallis test was performed comparing to Control; *p < 0.05; ***p < 0.001; ns, not significant.

Effect of PEG-PLA-TMZ nanomedicine on cell apoptosis

Previous studies have reported that TMZ alone increases mitochondrial membrane potential significantly (Yuan et al., 2021), our results demonstrate that PEG-PLA based TMZ nanomedicine further increases the mitochondrial membrane potential compared with TMZ itself (Fig. 5A). Besides, many researches tell that dysfunction of mitochondria produces large amount of ROS (Zorov, Juhaszova & Sollott, 2014). Therefore, we compared with the ROS levels between TMZ and TMZ nanomedicines. The results discover that nanomedicine significantly increased the ROS levels relative to TMZ itself. As a result, it induced more pronounced apoptosis (83.75%) compared with control group (2.82%) (Fig. 5C).

Figure 5 Effect of TMZ nanomedicine on the cell survival related processes.

(A) Effect of nanomedicine on mitochondrial membrane potential. The green fluorescence indicates a decreased mitochondrial membrane potential, and the red fluorescence indicates relative normal mitochondrial membrane potential. Scale bar = 20 μm. (B) Effect of nanomedicine on ROS levels (n = 5). Kruskal–Wallis test was performed comparing to Control; *p < 0.05; ***p < 0.001. (C) Effect of nanomedicine on apoptosis status by flow cytometry.

Discussion

Glioblastoma is the most common malignant tumor of the central nervous system. Previous studies have shown that chemotherapy drugs had been largely ineffective in treating glioblastoma, leading to suboptimal overall prognosis for patients (Schaff & Mellinghoff, 2023). In the study by Stupp et al. (2005), the combination of TMZ with concurrent radiotherapy followed by six cycles of temozolomide chemotherapy increased the median survival to 14.6 months, raised the 2-year survival rate to 27.2%, and achieved a 5-year survival rate of 9.8%. These results have led to increasing attention from clinical neuro-oncologists towards TMZ, establishing it as a frontline drug for treating patients with malignant gliomas. The current efficacy of temozolomide in the treatment of GBM remains below 45%, with the main limitations for clinical use including: (1) adverse reactions such as liver damage, gastrointestinal reactions, and bone marrow suppression after medication; (2) low effective drug concentration in the tumor area leading to insufficient anti-tumor effects (Poon et al., 2021). Furthermore, the stability of free TMZ in the cultured medium is relatively low, leading to rapid degradation and reduced efficacy. However, the controlled drug release technique using PEG-PLA-TMZ (PPT) significantly increases the stability of TMZ. This method extends the action time of TMZ, thereby enhancing its efficacy by maintaining a sustained release and preventing rapid degradation.

Ensuring the efficient aggregation and release of chemotherapy drugs in the tumor area has been a research hotspot in the field of cancer treatment in recent years (Ding et al., 2022). The low-toxicity drug delivery carriers prepared from materials can significantly reduce systemic adverse reactions of drugs while greatly improving treatment effectiveness (Li et al., 2010). Consequently, they can enhance post-chemotherapy survival quality for patients. The premise of targeted drug delivery is that polymer drug carriers exhibit strong stability in the bloodstream, effectively prolonging the circulation time of drugs in the body to achieve drug accumulation in tumor tissues. Early in vivo studies have shown that some PEG-polyester polymer nanomedicines experienced burst drug release in vitro release experiments, preventing complete drug delivery to tumor tissues, causing adverse reactions in normal tissues. Moreover, the accumulation of nanomedicines at tumor sites was mostly less than 5% of the injected dose per gram of tissue (Cabral & Kataoka, 2014). The possible reason for this outcome is that the structure of the prepared polymer nanoparticles is unstable, and the weak physical adsorption between the drug and the polymer is easily disrupted, leading to premature drug release.

In recent years, many researchers have been devoted to designing and synthesizing more structurally stable polymer nanocarriers through various modification methods, the therapeutic effects of anticancer drugs. Zhang et al. (2018) prepared PEG-PLA nanoparticles co-loaded with the anticancer drug paclitaxel (PTX) and the anti-angiogenic drug itraconazole (ITA) using a combination therapy approach. Compared to nanoparticles solely loaded with PTX, the introduction of ITA allowed for intermolecular interactions with PTX, which inhibits the drug’s crystallization ability within the nanoparticles and enhancing the stability of the polymer nanoparticles. The study demonstrated that these PTX/ITA co-loaded nanoparticles not only exhibited high anti-proliferative efficacy against non-small cell lung cancer cells but also improved the resistance of non-small cell lung cancer to PTX. Additionally, there was a significant enhancement in the accumulation of the nanoparticles in tumor tissues (Zhang et al., 2018). In order to achieve more structurally stable polymer nanoparticles, Lu et al. (2016) directly linked the topoisomerase I inhibitor 2-ethyl-10-hydroxy camptothecin SN38 to the polymer mPEG-PLA via ester bonds to obtain the polymeric macromolecular prodrug mPEG-PLA-SN38. The amphiphilic structure of mPEG-PLA-SN38 allows for self-assembly into polymer nanoparticles in aqueous solutions, overcoming the poor solubility issue of SN38. Furthermore, the compact structure of the polymer nanoparticles ensures their stability during transport in the body (Lu et al., 2016).

Cell apoptosis is a programmed cell death process that plays an important regulatory role in maintaining normal physiological states, developmental processes, and pathological conditions (Carneiro & El-Deiry, 2020). Reactive oxygen species (ROS) are a highly reactive group of oxygen-containing molecules, including superoxide anions (O2−), hydrogen peroxide (H2O2), hydroxyl radicals (·OH), etc (Cheung & Vousden, 2022). ROS are mainly generated from the mitochondrial respiratory chain, NADPH oxidase, cytochrome P450 enzymes, etc (Chen et al., 2021). These enzyme systems maintain the redox balance within cells under normal circumstances. When the cellular environment changes, the activity of these enzyme systems increases, leading to an increase in ROS production. On the other hand, there are antioxidant systems within cells, such as superoxide dismutase (SOD), glutathione peroxidase (GPx), glutathione-S-transferase (GST), which can eliminate ROS, maintaining the redox balance within cells. When ROS production is excessive or the clearance system is inhibited, ROS accumulates in cells, triggering oxidative stress, which in turn promotes cell apoptosis (Wang et al., 2020). In this study, we discovered that TPPT induced more production of ROS compared with TMZ alone, which is why intensity of apoptosis is increased.

Those results provide a feasible material (PEG-PLA) to be used as a carrier to improve the efficacy of TMZ, our experimental results indicate a significant enhancement in the therapeutic efficacy of TMZ after encapsulation in PEG-PLA, suggesting that this strategy provides a feasible approach to address the existing issues with TMZ. However, so far, only cell-based in vitro experiments have been conducted. Further animal studies and exploration in clinical stages are required to validate these findings.

Conclusions

This study aimed to improve the efficacy of TMZ in treating glioblastoma by using PEG-PLA nanocarriers. The encapsulation of TMZ in PEG-PLA significantly enhanced its therapeutic efficacy, as demonstrated by increased ROS production and cell apoptosis compared to TMZ alone. This suggests that PEG-PLA is a promising carrier for TMZ, addressing its current limitations of adverse reactions and low effective concentration in tumor areas. Future studies should focus on validating these findings in animal models and exploring clinical applications. The potential of PEG-PLA to enhance drug delivery and reduce systemic toxicity warrants further investigation. These results provide a promising strategy for improving TMZ-based therapy, contributing to more effective treatment options for glioblastoma patients.

Supplemental Information

Supplemental Information 1 Raw data.

Supplemental Information 2 Figure 5C Flow Cytometry Data.

Effect of nanomedicine on apoptosis status by flow cytometry.

We thank Michael Zhu for his assistance in editing the English of this manuscript.

Additional Information and Declarations

Competing Interests

Author Contributions

Data Availability

The authors declare that they have no competing interests.

Jun Yu conceived and designed the experiments, performed the experiments, analyzed the data, prepared figures and/or tables, authored or reviewed drafts of the article, and approved the final draft.

Yonghua Xue conceived and designed the experiments, performed the experiments, authored or reviewed drafts of the article, and approved the final draft.

The following information was supplied regarding data availability:

The data is available at figshare: Xue, Yonghua (2024). Flow Cytometry Data. figshare. Figure. https://doi.org/10.6084/m9.figshare.26064226.v1.

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
