# Peer review of "Preparation of transferrin-targeted temozolomide nano-micelles and their anti-glioma effect"

_PeerJ, doi:10.7717/peerj.17979_

## Round 0.1 · original submission · Minor Revisions

The review comments are positive on the manuscript. Please address all questions sufficiently.

Reviewer 1 ·

Basic reporting

The manuscript is written in clear and unambiguous professional English throughout. Literature references provide sufficient background and context for the field. The article structure, figures, and tables are professional, and the manuscript is self-contained with results that are relevant to the hypotheses. However, there are some minor errors and inconsistencies that need to be addressed.
- Abbreviations that appear for the first time should be accompanied by their full names. The absence of full names for "PP," "PPT," and "TPPT" may cause misunderstanding.
- A grammar error in line 73. "on one hand" should be "on the one hand."
- Lines 73-76 ("On the one hand … enhancing bioavailability"), lines 194-195 ("Besides … ROS"), and lines 200-202 ("Previous studies … for patients") need references.
- The authors should provide a clear background on the components of the drug delivery system, including their functions, advantages, and the reasons for choosing that system.

Experimental design

The research question is well-defined, relevant, and meaningful. The research addresses an important gap in the treatment of glioblastoma. However, some methods lack sufficient detail..
- If the authors do not specify the distinct meanings of "PPT" and "TPPT," it may lead to misunderstandings. In Figures 3A and 3B, "TPPT" means "PEG-PCL-TMZ," but in other parts of the manuscript, it may indicate TRF-loaded "PPT."
- The statistical methods are not clearly explained. The authors should correspond each statistical method to the relevant figure in the figure legends.
- In Figure 3A, "TPPT" remains the same size over 30 days. Does this mean TMZ cannot be released from "PPT" given the size differences between "PP" and "PPT" in Figure 2B?
- Please add the methods used in experiments for Figures 3B and 4B and explain the significance of the results.
- In Figure 3B, clarify why PTX release is detected instead of TMZ.
- Explain the meaning of "C" in Figure 4A and "CTMZ" in Figure 4B. Do they share the same meaning?
- Add scale bars to Figures 2A and 5A.

Validity of the findings

The validity of the findings is supported by robust data, but the manuscript would benefit from a more explicit discussion on the impact and novelty of the findings. It is important to note that the in vitro experiments were conducted using only one cell line, and the absence of in vivo experiments limits the rigor of the results.

Additional comments

The manuscript would benefit from a thorough edit and get rid of any irregulars and misunderstandings. While the study is innovative, a more explicit discussion on the impact and novelty of the findings compared to existing therapies would be beneficial. Including more recent studies in the literature review could provide a better context for the current research and demonstrate how it advances the field. Ensure that all figure legends are comprehensive and provide sufficient detail to be self-explanatory. Additionally, enhancing the clarity and quality of some images would improve the presentation of the data.

·

Basic reporting

PASSED: Literature references, sufficient field background/context provided.
PASSED: Professional article structure, figures, tables. Raw data shared.
PASSED: Self-contained with relevant results to hypotheses.
FAILED: Clear and unambiguous, professional English used throughout.

Critical comments 1:
The rationel of "TRF targeted" was not clearly addressed.

Critical comments 2:
Abstract should be re-written to make it more concise and clear.

Critical comments 3:
Line 183-189, page 13: Readers may not easily recognize which are those bars you are describing. What does "PEG-PLA itself" refer to the bars in Figure 4A, "C" or "PP" (must this one, but need to make it more clear)? Likewise, how about the "after loading TMZ" and "after loading TRF"? I guess they are "PPT" and "TPPT", respectively. Was that right? Presumably, in this situation, a significance label as indicated (Figure 4A, annotated PDF while reviewing) should be considered. Besides, it seems that descriptive main text in "Effect of Copolymers on the cell viability of U251" (Line 183, page 13) for Figure 4B is missing.

Critical comments 4:
[For Figure 1] The full name of all abbreviations should be clearly defined or prompted, otherwise the readers may not understand. Brief descriptions for each process should be intently texted for panel A (self assembly) and B (drug loading).

Once taking all these lines together ---- written context in "Synthesis of nanomedicines", "Characterization of morphology, particle size, zeta potential and stability of nanomedicine" in Methods section, and "In order to...conjugate." (164-169, page 12) in Results section ---- I think you authors mainly wrote for Figure 1B while excluding Figure 1A. It is highly recommended to write for it and revise the relevant in Methods sections as mentioned above.

Experimental design

PASSED: Original primary research within Aims and Scope of the journal.
PASSED: Research question well defined, relevant & meaningful. It is stated how research fills an identified knowledge gap.
PASSED: Rigorous investigation performed to a high technical & ethical standard.
FAILED: Methods described with sufficient detail & information to replicate.

Critical comments 5:
Please move "92 Cell culture and reagents" to the very end of "116 Drug loading and release performance of targeted nano carrier" to fit the chronological order.

Critical comments 6:
We would like you authors to combine "98 Synthesis of nanomedicines" and "108 Characterization of morphology, particle size, zeta potential and stability of nanomedicine" as one section tittled "Synthesis and characterization of nanomedicines" to reduce the existing redundancy. This is a recommendation.

Critical comments 7:
Line 164-169, page 12: Methodology details should go to Methods section. By the way, what is the "synthesized copolymer"? Does the polymer contain drug TMZ? If it doesn't, is that NH2-PEG-PLA (Line 99, page 9)? So, when TMZ drug soultion was introduced into the aqueous system to faciliate the formation of TMZ loaded PEG-PLA micelles?

Validity of the findings

PASSED: Impact and novelty not assessed. Meaningful replication encouraged where rationale & benefit to literature is clearly stated.
PASSED: All underlying data have been provided; they are robust, statistically sound, & controlled.
PASSED: Conclusions are well stated, linked to original research question & limited to supporting results.

No other comments.

Additional comments

Critical comments 8:
TMZ, as a first-line drug, has less endured effect on GBM bearing only ~45% efficacy, but surprisely it has high permeability and nearly 100% bioavailability. Why? Because of a super rapid elimination with much smaller half-life? Could you authors please explain more in Line 68-69, page 7?

How about the stability of free drug TMZ in cultured medium? PEG-PLA (PP) micelles are bio-inactive in regarding GBM. Controlled drug release using PEG-PLA-TMZ (PPT) technique does significantly increase efficacy of TMZ. Does the increased stability of TMZ using this technique extend the action time and thereby enhance its efficacy?

If these were well addressed, in my viewpoints, the hypothesis as claimed in this manuscript (Line 89-90, page 8) would stands more explictly.

Critical comments 9:
Other omitted major/minor comments for revision are available in annotated PDF file.

---

## Round 0.2 · accepted · Accept

Authors have addressed all of the reviewers' comments. This manuscript is ready for publication.

Reviewer 1 ·

Basic reporting

The author made appropriate revisions to the article according to the comments, corrected obvious language errors, and cited literature reasonably. The article is well-constructed and well-written.

Experimental design

The experimental method of the paper has been supplemented and modified to ensure repeatability. Both the experimental method and the legends are described in detail.

Validity of the findings

The drug delivery system's application has improved the drug's effects. As a preliminary exploration, good results have been obtained. The author should add validation dimensions in subsequent studies to provide more reliable research results.

Additional comments

The author has made comprehensive revisions to the article to avoid language that could cause confusion and misunderstanding and to enhance the readability of the images. The article is now more readable than before.

·

Basic reporting

No comments.

Experimental design

No comments.

Validity of the findings

No comments.

Additional comments

Suggest to accept this manuscript for publication.